# Comparing Friends and Peer Tutors Amidst COVID-19 Using Social Network Analysis

**Nurul Zahirah Abd Rahim** [1,2], **Nurun Najwa Bahari** [1], **Nur Syaza Mohd Azzimi** [1], **Zamira Hasanah Zamzuri** [1], **Hafizah Bahaludin** [3], **Nurul Farahain Mohammad** [4] **and Fatimah Abdul Razak** [1,*]

1. Department of Mathematical Sciences, Faculty of Science & Technology, Universiti Kebangsaan Malaysia (UKM), Bangi 43600, Selangor, Malaysia
2. Department of Mathematics, College of Computing, Informatics and Media, Universiti Teknologi MARA (UiTM), Melaka Branch, Jasin Campus, Merlimau 77300, Melaka, Malaysia
3. Department of Computational and Theoretical Sciences, Kulliyyah of Science, International Islamic University Malaysia, Kuantan 25200, Pahang, Malaysia
4. Kampung Oren, Ulu Tiram, Johor Bahru 81800, Johor, Malaysia
* Correspondence: fatima84@ukm.edu.my

**Abstract:** COVID-19 has drastically changed the teaching patterns of higher education from face-to-face to online learning, and it has also affected students' engagement socially and academically. Understanding the nature of students' engagement during online learning can help in identifying related issues so that various initiatives can be implemented in adapting to this situation. In this study, social network analysis is conducted to gain insights on students' engagement during COVID-19. Directed and weighted networks were used to visualize and analyze friendship as well as peer tutor networks obtained from online questionnaires answered by all students in the class. Contrasting friends and peer tutors reveals some hidden interactions between students and shines some light on dynamics of the online learning community. The results indicate that, popular and important peer tutors may not be high achievers and thus possibly contributing to the spread of misinformation in the online learning community. By comparing weighted indegree and betweenness centrality values, we suggest approaches to cultivate a healthy online learning community. This study highlights the use of social network analysis to assist and monitor students' engagement and further formulate strategies in order to make the class a conducive online learning community, particularly in the advent of online learning in higher education institutions.

**Keywords:** student engagement; friendship networks; peer tutor networks; COVID-19; online learning community; higher education

**MSC:** 05C90; 05C82; 68R10





## 1. Introduction

Due to the COVID-19 pandemic, some learning processes had to change dramatically. Educators employ various strategies to adjust to these changes. As an alternative to the previous face-to-face learning techniques, there are several online learning tools available today [1]. Online education is not a new idea [2], it used to be an option or substitute for a face-to-face class. The integration needs to be given serious consideration to ensure not only the continuation of learning but also the assurance of quality learning by creating a better learning community, especially when there are limitations on conducting face-to-face classes.

One of the key components of a learning community is students' engagement. Students' engagement not only helps students perform better academically [3] but also helps students develop their personalities [4]. Moreover, this engagement is vital to build an online learning community that advances knowledge [5]. The engagement involves three types of interactions, peer-to-peer interaction [6,7], student interaction with teachers [8],

and student interaction with the learning content [9]. Peer-to-peer interaction provides mutual advantages for all participants and helps to improve understanding, self-confidence, and forming critical and creative thinking [10]. According to [11], peer-to-peer interaction is equally or more significant than the traditional teaching approach. The achievement of students may be affected by this absence of interaction [12].

Involvement in friendships and peer tutoring are two crucial forms of engagement that are frequently linked to interactions between students. Both kinds of connections between students develop naturally. Previous studies highlight the importance of friendships amongst higher education students in the learning environment. Ref. [13] focuses on the role of friendship in the online learning environment. Ref. [14] emphasizes the value of developing friendships among students to support their learning and foster an atmosphere where everyone is valued as a source of knowledge. Friends tend to be people with similar attributes (also known as homophily) such as gender and religion. This inevitably impacts the structure of a learning community [15]. Previously, in [16,17], we studied the relationship between peer tutoring and student performance. Ref. [17] highlights that peer tutoring does not necessarily correspond with academic success. Therefore, we take into account other variables to further understand peer tutoring and academic performance, including homophily, which can be captured using friendship networks.

This study emphasizes social network analysis as the primary technique for examining the dynamics of the formation of an online learning community. A practical solution can be recommended to educators to help monitor and create a healthier learning environment. This study is a continuation of [16,17], further using centrality measures to understand the learning community of Malaysian students. Due to the dearth of studies related to online learning communities, particularly in Malaysia, it is a field that requires attention, especially when dealing with crises such as the COVID-19 pandemic.

Social network analysis is an approach that utilizes data mining to investigate the social structure of a community via graph theory. This strategy has been successfully applied in a variety of disciplines, including finance [18,19], scientific collaboration [20,21], and education [16,17]. In the context of education, this approach can be utilized as a benchmark for monitoring student association through networks and developing solutions that are pertinent to concerns and challenges [22–24]. This enables educators to build a better learning community by ensuring that students' engagement results in outcomes that are beneficial to the students.

One goal of this study is to encourage researchers, especially educators, to utilize social network analysis in order to investigate the structure of online learning communities. The main contribution of this study is to highlight the importance of contrasting friendships and peer tutor relationships to reveal the dynamics of the formation of an online learning community using the social network analysis approach. This study focuses on undergraduate students in Malaysia with online learning experience throughout the whole semester due to movement restrictions during the pandemic COVID-19. To ensure that data are gathered properly, the use of technology and concerns regarding data collection have been taken into consideration. Section 2 discusses relevant literature and Section 3 provides detailed explanations of our research methodology. Findings and analyses are presented in Section 4. Section 5 discusses the findings and offers potential strategies to encourage the formation of a healthy online learning community.

## 2. Related Works

Students' interaction is important to improve understanding, self-confidence, and forming critical and creative thinking [10]. Two types of interactions that are often associated with relationships between students are friendships and peer tutors. When two or more people wish to interact with one another, a friendship can develop because of their shared values, common interests, mutual respect, and capacity for emotional support [25]. In the process of peer tutoring, students work in groups or pairs to directly support one another's learning. Peer tutoring could be more beneficial for students who are academ-

ically driven than the formal tutoring programs offered by the university [26]. Ref. [27] discovered that students with similar friendship groups have similar levels of achievements in the classroom. This shows the existence of the homophily effect in the community. If the tutor has a strong academic record, the benefits of peer tutoring can be immense. The chosen peer tutors, however, are occasionally not high achievers [16], which may indicate less or sometimes incorrect understanding of the educational content.

Student achievement is commonly associated with peer tutoring relationships. Based on physical interactions, ref. [12] investigated peer learning relationships in the learning community among Malaysian students. The study's findings revealed that the learning community refers to top achievers as role models. For the benefit of the learning community, ref. [12] suggested that these students need to be groomed as peer leaders. A qualitative study was carried out by [13] to examine how undergraduate students form an online learning community. Ref. [13] discovered a connection between the sense of belonging in the classroom and friendships among the students. Therefore, friendship is an important factor contributing to the peer tutoring relationship towards developing a learning community.

The gathering and analysis of student relationship data is required in order to comprehend the effects of friendships and peer tutor relationships on the development of the learning community. The collection of this data is important in understanding the structure of the community formed. Ref. [9] used discussion data between students in a forum to explore the influence of student in a network. Ref. [28] provided an introduction and guideline to use social network analysis in order to analyze the structure of community in the classroom. On the basis of graph analytics modeling, ref. [12] further provided a framework for investigating students' physical interaction in peer learning. These studies can serve as a roadmap for more readily picturing and comprehending the community structure formed using the social network analysis approach. Compared to the previous studies, this study highlights the importance of considering both friendship and peer tutor relationship in the online learning community using social network analysis.

Social network analysis is a method that utilizes graph theory and statistics to understand the structure of social networks. Nodes and edges are the main components in building a network. In the context of student-to-student interaction, nodes are represented by students and edges are represented by interaction or relationships that exist between students. To identify the structure and important players in a group, social network analysis investigates the social ties inside the group [21]. Various methods of social network analysis, called centralities, are used to understand the nature of the social network formed by students [12,16,17,29].

Most studies explain the popularity of students in a social network using centralities [7]. Indegree and betweenness centrality are two common forms of centrality measures employed by numerous researchers. The most popular students in the community can be determined in part by looking at indegree centrality. Refs. [16,17] used indegree centrality to examine how peer tutoring affected students' academic performance. Based on the results of these two studies, some low achievers became the main reference partner in the learning community. Ref. [17] claimed that popularity as peer tutor and academic performance are not always correlated. Ref. [12] identified peer leaders based on the students' physical interaction in peer learning community. This study used social network analysis to visualize the learning community and successfully identified potential peer leaders from each community. Ref. [12] found that only excellent students became the most influential students in the community. However, these findings contradict findings in [16,17].

Betweenness centrality is a global measure that can provide an overview of the characteristics of the nodes in the entire system. For example, a student may have a high betweenness centrality because he or she is an important person in the network by connecting two groups of students [30]. This concept is also important in understanding the structure of the student network and the dissemination of information to the entire network. Ref. [31] used interactions that occurred in university residences to illustrate the spread of COVID-19 among students if standard operating procedures were not followed. This shows

that network analysis can assist in strategy development and understanding information transmission in the community. Ref. [32] highlighted the importance of monitoring specific individuals in controlling the COVID-19 outbreak using a student friendship network. In comparison to separating individuals with a high degree centrality or high betweenness centrality, ref. [32] indicated that isolating individuals with the highest betweenness centrality was more effective at slowing and halting the spread in the community, thus flattening the infection curve faster. This is because the highest betweenness centrality individuals serve as connectors of different groups in the community. In addition, ref. [33] used a weighted student friendship network to prove that monitoring individuals with the highest betweenness centrality values who act as super-spreaders can be used in predicting the worst-case scenario and simulating the super-spreading dynamics of COVID-19. It brings to light the utility of network analysis in introducing a better monitoring approach by considering the role of individuals in the spread of epidemics in a community. Therefore, monitoring targeted individuals helps in significantly reducing infection rates compared to monitoring random individuals.

This demonstrates the effectiveness of network analysis in simulating, forecasting, and containing the spread of disease within a society. In the spread of an epidemic, the main goal is to slow down, reduce, and prevent the spread from continuing. This motivates our study by considering the role of students in encouraging the 'spread of knowledge' and preventing the 'spread of misinformation' in an online learning community. Monitoring influential students who are 'spreaders' in the network is not only essential for the spread of correct knowledge but also for ensuring that knowledge development continues to take place in the learning community.

In social network analysis, visualization is important to understand and illustrate the whole structure of the network. Various software programs have been used to visualize and analyze social network data, such as igraph package in R software [12,34], NodeXL Software [35], Gephi [17,36], and Ucinet [30].

Based on these studies, social network analysis can help in understanding the problems that arise from the interaction between students. Moreover, it can be used in predicting student performance, in formulating strategies to encourage or prevent spreads of information (or disease) and also provide an overview of the interactions that occur. By identifying these interactions, educators may be able to identify obstacles to the formation of an effective online learning community and the proper dissemination of knowledge.

## 3. Data and Network Visualization

The process for gathering and analyzing the data for this study is covered in this section. Data collection is the first step in this process, followed by data management, data visualization, data measurement, and software used.

### 3.1. Data Collection

This study involved 41 students from Universiti Kebangsaan Malaysia. Due to COVID-19 induced movement restrictions, the class was conducted online over the whole semester. An online survey was distributed as it was the most convenient way to reach and engage with the respondents. The research objectives, the purpose, the confidentiality of the data obtained, and other information related to the research were stated in the questionnaire. This study did not involve any physical, physiological, or social risks to students. All students who participated in this survey agreed to be directly involved in this study.

### 3.2. Questionnaire

The questionnaire from [17] was adopted in this study. It was converted to an online Malay language questionnaire to improve the data collection and analysis. Students were required to list ten names of their close friends and ten names of their peer tutors in the same class in descending order of importance. The order of the name on the list starts from the most important to least important relationships. The weight of the edges was

assigned based on the order of the name filled in by students in the questionnaire. We acknowledge that, although this information was provided voluntarily by the students, there is a possibility that it was not stated accurately and truthfully.

When selecting which online form to be used, we considered usability. The customization and versatility of Jotform's questionnaire creation tools simplify the process of designing an online survey for this study while also helping to prevent the issues of students filling up the form untruthfully due to laziness. Ref. [7] suggested that the list of every student enrolled in the course should be provided in the survey to make it easier for students to respond. The process of entering friends' names could be challenging, especially if the student is unable to recall their friend's name, and it might be more challenging if a big number of students are registered in the class they attend. Thus, we chose to use Jotform as the online questionnaire platform due to its autocomplete feature. When students begin typing the names of their friends, as seen in Figure 1, the autocomplete feature in Jotform is intended to speed up searches. This helps to provide ideas to respondents, remove lengthy choice lists, and make it easier for respondents to complete forms. Students were required to fill in every space and could not leave any spaces blank. The 'no friends' option is available in all of the answer boxes in this section for students who do not have any friends.

**Figure 1.** An English version of the distributed online Malay language questionnaire with the autocomplete feature in Jotform.

*3.3. Network Visualization*

Using data from the questionnaire as a starting point, the following approach was taken to create weighted and directed networks for friendship and peer tutor relationships. A graph or network is defined as,

$$G = (V, E) \tag{1}$$

formed by a set of nodes, $V$ and a set of edges, $E$. The edges in our study are directed and weighted. Any object that may be connected to another object by a relationship is considered to be a node, and the connections between nodes are known as edges. $N$ represents the number of nodes and $L$ is the number of edges that exist in the network.

In our networks, students are represented as nodes, and edges are represented as students' connections in the classroom community. To protect students' privacy, each student was assigned a unique ID that was used as a node label as in Figure 2. The edge weight was assigned based on the order of the name listed in the online questionnaire (Figure 1). For example, node 1 nominates node 2 as the first close friend while node 3 is the second close friend. Then, the edge weight between nodes 1 and 2 is assigned as a

weight of ten, while the edge weight between nodes 1 and 3 is assigned as a weight of nine. In other words, the first name on the list received a weight of 10, the second name received a weight of 9, and so on. Node 1 is referred to as the source, and nodes 2 and 3 are referred to as the targets. This depicts the direction of the edges from source, $i$ to target, $j$ with the edge weight, as illustrated in Figure 2.

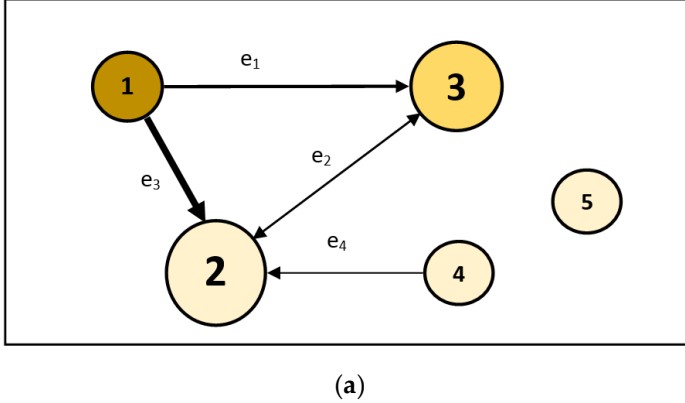

| Node | 1 | 2 | 3 | 4 | 5 |
|------|---|----|---|---|---|
| 1 | 0 | 10 | 9 | 0 | 0 |
| 2 | 0 | 0 | 8 | 0 | 0 |
| 3 | 0 | 8 | 0 | 0 | 0 |
| 4 | 0 | 8 | 0 | 0 | 0 |
| 5 | 0 | 0 | 0 | 0 | 0 |

(**a**)　　　　　　　　　　　　　　　　　　　　　　　(**b**)

**Figure 2.** (**a**) Weighted and directed network constructed from a questionnaire filled by five students. The size of the node represents its weighted indegree value and the thickness of the edge represents the edge weight. The highest achiever (highest marks) is represented by the darkest node, while the lower achievers are represented by the lighter colored nodes, respectively. (**b**) The adjacency matrix, $A_{ij}$, representing the network constructed from the student network in (**a**).

A weighted and directed network, $G$ can be mathematically described using an adjacency matrix, $A_{ij}$, such that $i, j \in V(G)$. The direction of the edge points from $i$ to $j$ if student $i$ reports an interaction with student $j$. The size of $A_{ij}$ is $N \times N$. Figure 2 shows a network of five students and it is the corresponding adjacency matrix of size $5 \times 5$. In general, the adjacency matrix of the network in this study, $A_{ij}$, is defined as,

$$A_{ij} = \begin{cases} a_{ij} & \text{if there is an edge pointing from } i \text{ to } j \\ 0 & \text{otherwise} \end{cases} \tag{2}$$

where $a_{ij}$ is the value obtained based on the edge weight. It should be noted that this relationship is not symmetrical; if student $i$ reports an interaction with student $j$ but student $j$ does not report any connection with student $i$, then the direction of the edge points from $i$ to $j$. The edge is recognized as bidirectional or reciprocal if both students indicated that they engaged with one another. For example, in Figure 2, the only reciprocal relationship is between nodes 2 and 3. This is the defining feature of a directed network; its connections are asymmetrical. The edge's weight serves as an indicator of the intimacy of students' relationships. The size of nodes indicates centrality values, with larger nodes having higher values. The node's color displays the total marks (achievement) obtained by each student in their course.

### 3.4. Network Measurements

Edges are used to quantify the level of engagement within the community through centrality. Each centrality measure brings a different interpretation to the network. This study considers two centrality measures, which are weighted indegree centrality and betweenness centrality as well as their relationship to students' performance on the subject. To evaluate the difference between friendship and peer tutor networks, this study reports the average degree, density, clustering coefficient, and average path length.

A weighted and directed network of students' interactions is considered. Degree is a measure that counts the number of edges connected to a vertex (student). For every node $i \in V$, we denote the weighted indegree by $k_i^{in}$, representing the weight of all edges pointing towards a student. It is calculated using Equation (3) [29]. If many students nominate student $i$ as their top friend or top peer tutor, resulting with many edges pointing

to student $i$, then student $i$ will have a high value of $k_i^{in}$. It is normally considered a measure of popularity in a social network.

$$k_i^{in} = \sum_{j=1}^{N} A_{ij} \tag{3}$$

The shortest path is used to calculate betweenness centrality. It represents the minimum number of edges that connect two students in a network. Betweenness centrality indicates how often a student lies on the shortest path between two other students. The edges of the network are considered as undirected and unweighted for the calculation of betweenness centrality. In the context of this study, high betweenness centrality students act as bridges between two or more connected groups of students. This position is important in the matter of information transfer to the whole network. Refs. [37,38] defined betweenness centrality as in Equation (4). Let $\sigma_{st}$ be the number of shortest paths from $s \in V$ to $t \in V$, and $\sigma_{st}(v)$ denotes the number of shortest paths from $s$ to $t$ that some $v \in V$ lies on. Ref. [39] has further improved and proposed an algorithm for the betweenness centrality measure which we utilize through Gephi.

$$C_B(v) = \sum_{s \neq v \neq t \in V,} \frac{\sigma_{st}(v)}{\sigma_{st}} \tag{4}$$

In the context of student networks, the average degree, $\langle k \rangle$, provides an idea of the number of friends a student might have [29]. In general, it can be used to measure the number of edges compared with the number of nodes. It can be obtained by dividing the number of edges by the number of nodes denoted in Equation (5).

$$\langle k \rangle = \frac{L}{N} \tag{5}$$

To obtain the network density of a graph, $\rho$, the total number of edges in a graph is divided by the total number of possible edges, $N(N-1)$. Mathematically, it can be formulated as in Equation (6). In the student networks, higher density values represent a denser connection among students.

$$\rho = \frac{L}{N(N-1)} \tag{6}$$

The clustering coefficient (also known as transitivity) measures the likelihood that two friends of a student are also connected directly. The clustering coefficient is defined as the ratio of the number of closed triangles to the number of all connected triplets. The average path length of a graph, $l$, is defined as the average number of edges along the shortest path for all possible pairs of network nodes and is represented using Equation (7). $d_{ij}$ is the distance between nodes $i$ and $j$ for all pairs of nodes in the network.

$$l = \frac{1}{N(N-1)} \sum d_{ij} \tag{7}$$

Connected components represent the set of nodes in a network that are connected to each other by at least one edge. In this study, the network consists of one giant connected component which implies that all students are connected and communicate with each other. A disconnected component in a network display either isolated students that are not connected with any other students or the existence of completely separate communities.

*3.5. Software*

In this research, the open source software Gephi was utilized to visualize and analyze the data. The position of each node in the figures was fixed so that changes in edges can be compared for friendship and peer tutor networks. Two Excel files (csv file) were prepared. The first file contains three columns, which are the source, target, and weight. For example, if student $i$ nominates student $j$ as the closest friend, then $i$ will be in the source column, and $j$ is in the target column with a weight of 10 to represent the intensity of the relationship. The second file contains the properties of the nodes, such as the student's performance

on the subject. The result of this study uses the calculation of average degree, network diameter, density, clustering coefficient, and average path length from the statistics panel. Fruchterman–Reingold layout is a force-directed layout algorithm which treats edges like springs that move nodes closer or further from each other. This layout was chosen to be applied to the network. Node color is chosen to represent students' performance and node size represents centrality values.

## 4. Results

This study provides a comparative analysis based on the structure of friendship and peer tutor networks. In particular, network topology was reported, such as average degree, network diameter, density, clustering coefficient (also known as transitivity), average path length, centrality measure, weight indegree, and betweenness centrality. This analysis aims for a better understanding of the dynamics within the online learning community based on friendship and peer tutor interactions between students during the COVID-19 pandemic. Furthermore, we suggest recommendations to tackle the concerns raised regarding the learning community in the next section.

Visualizing the network is frequently the first step in social network analysis. Based on the network diagrams, existing assumptions can be identified and new hypotheses can be generated. The network diagrams that depict the online learning community were visualized. Each node represents a student and an edge connecting two nodes denotes a relationship that exists between two students; in this research, the relationship is either friendship or peer tutoring. When a student refers to a peer tutor or lists another student as a friend, the edge is pointing in that direction, and the edge weight indicates the strength of the relationship. The weight of the edges was determined by the order in which a student's name was entered on the online questionnaire as explained in Section 3.3.

### 4.1. Comparing the Network Topology of Friendship and Peer Tutor Networks

Table 1 provides an overview of the friendship and peer tutor network of the 41 students (a whole class) participating in this study. Peer tutor networks have 20% fewer edges than friendship networks. The connected components being 1 indicate that all students are connected in terms of friendships and peer tutors. This means that during the COVID-19 pandemic, students at least have one friend or one peer tutor friend in the class while learning online. This study shows that there is interaction between students during the pandemic, however, there may have been a decline as suggested by [40]. Our study shows that all students are still connected in the community, even though online learning was implemented due to COVID-19. On average, a student has more friends—up to eight close ones—compared to peer tutors. A student only has six peer tutors on average. This shows students of this class engaged less as peer tutors. Lack of engagement among students pertaining to academic matters is not surprising.

**Table 1.** Topological summary of friendship and peer tutor networks.

| Network Measures | Friendship Network | Peer Tutor Network |
|---|---|---|
| Nodes, $N$ | 41 | 41 |
| Edges, $L$ | 344 | 274 |
| Average degree, $\langle k \rangle$ | 8.39 | 6.683 |
| Average degree (weighted) | 48.829 | 43.146 |
| Density, $\rho$ | 0.21 | 0.167 |
| Clustering coefficient | 0.583 | 0.487 |
| Average path length, $l$ | 2.39756 | 2.807175 |
| Connected component(s) | 1 | 1 |

The average path length of the friendship network is shorter than the peer tutor network. The fact that there were more friendships than peer tutoring relationships shows that students were more sociable and less academically engaged. This can be explained by the higher number of edges in the friendship network and the existence of edges

between students that can make the shortest distance between the two furthest nodes in the friendship network smaller compared to the peer tutor network.

The clustering coefficient measures the tendency of nodes in a network to group together. In this study, the clustering coefficient for the friendship network is higher than the peer tutor network. The higher number of edges in a friendship network results in a higher tendency of students to cluster together (a group of three students). The clustering coefficient in the peer tutor network is 16% lower than in the friendship network. Compared to the 20% fewer edges, this implies that the clustering structure in the friendship network is still somewhat intact in the peer tutor network.

### 4.2. Friends and Peer Tutors

Weighted indegree centrality is used to determine the popularity of a student among their classmates. Figure 3 shows that the weighted indegrees of the friendship and peer tutor network are highly correlated to each other with a coefficient of correlation of 0.798. From the scatter plot, most of the points lie close to the straight line. In general, this demonstrates that students who are proclaimed by many as friends in the class also serve as peer tutors in the same class. This indicates the importance of friendship as highlighted by previous research, especially in the contribution of friendship to the formation of an online learning community. Your friend is likely to be the person you will ask for academic help.

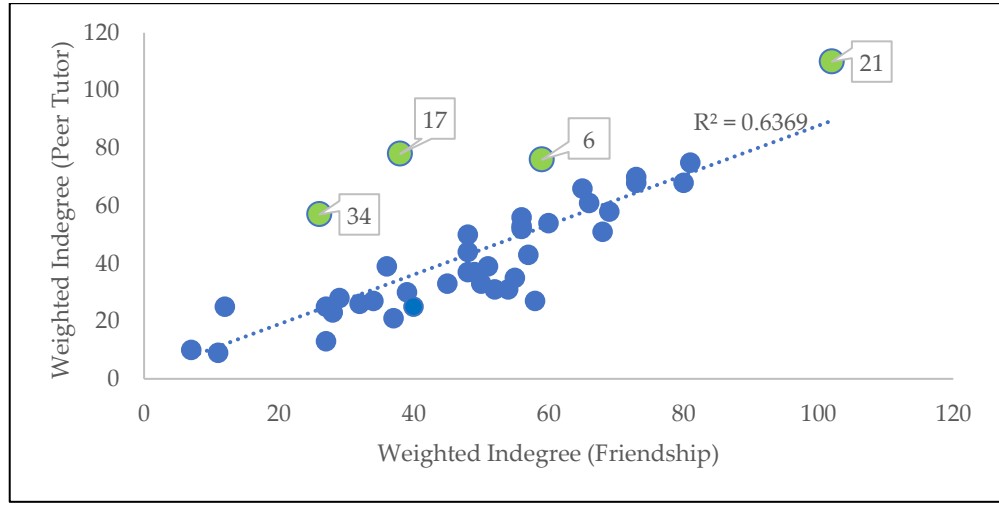

**Figure 3.** The relationship between weighted indegree of friendship network and weighted indegree of peer tutor networks. The green nodes represent outliers in the scatter plot.

There are four outliers (green nodes) in Figure 3 and they are nodes 6, 17, 21, and 34, representing four students with relatively high weighted indegree in the peer tutor network compared to the friendship network. These outliers are larger in the peer tutor network but smaller in the friendship network in terms of node size, as seen in Figure 4. These students tutor other students in the class more than they do other friendly activities. They are willing to help and perhaps able to teach other students in the community. Therefore, they are important for knowledge dissemination in the online learning community, even though some of them may not have many close friends.

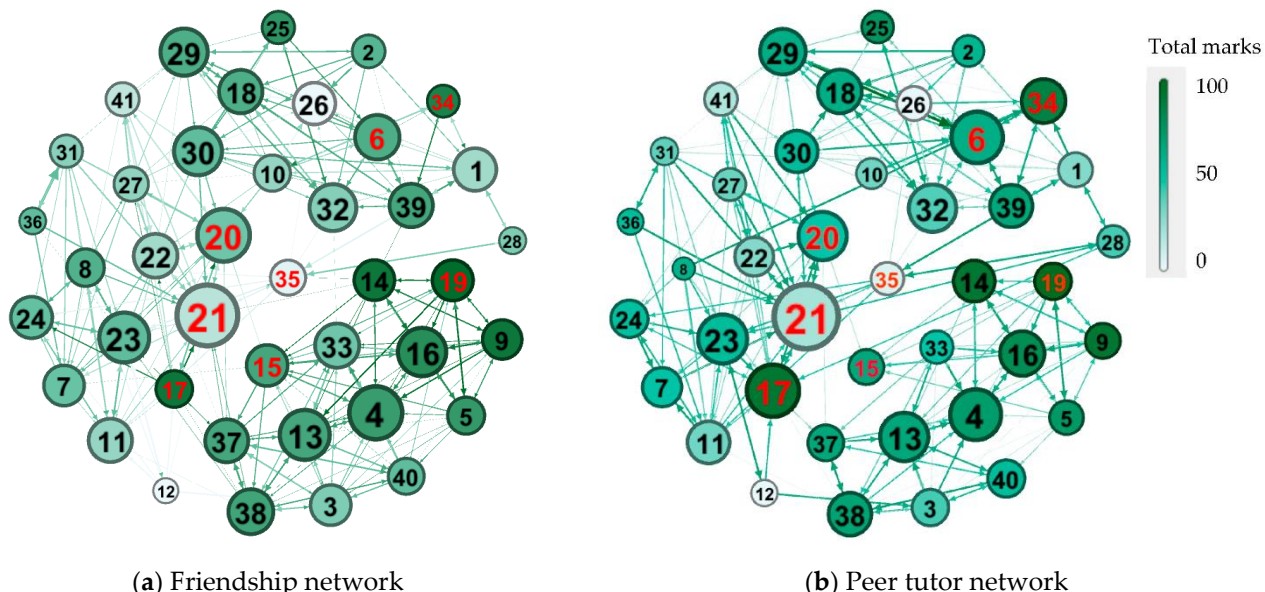

(**a**) Friendship network          (**b**) Peer tutor network

**Figure 4.** The size of the node represents weighted indegree and its color represents the total marks. Larger nodes represent higher weighted indegree and darker colored nodes represent higher marks. Red font is used to highlight the nodes that will be discussed in this section.

In Figure 4, nodes 17 and 34 are high achievers while node 6 is a moderate achiever. Nodes 17 and 34 obtained significantly higher marks than the marks of their peer tutor (Figure 5). In addition, nodes 17 and 34 were referred to as peer tutors because they were known as excellent students from their past achievements. The results of this study are similar to the study from [12] where high achievers become the important students in the learning community. This may be the reason why these students are referred to as peer tutors by other students, although they may have relatively less friends. Node 17 is a good achiever but is from the minority gender. Hence, this might be the reason that this student has not been listed as close friends by many students.

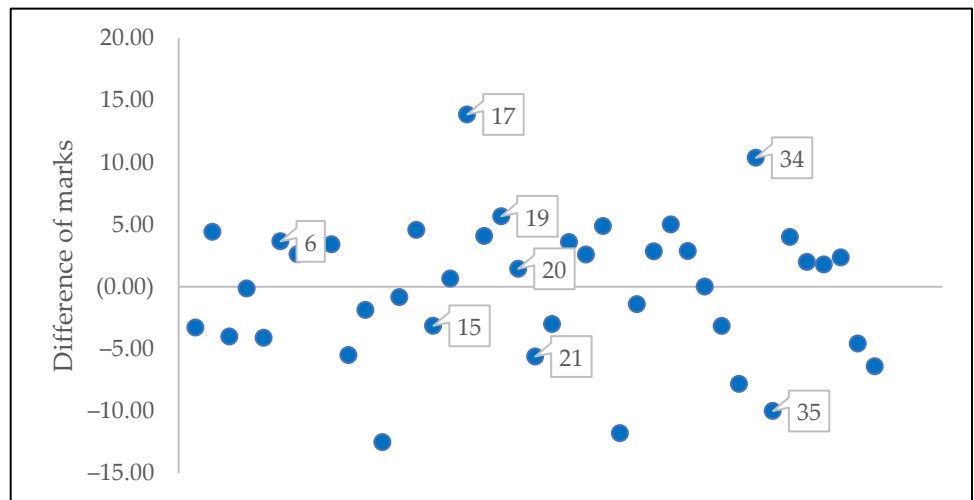

**Figure 5.** The difference of marks between the student and the student's peer tutors.

In this study, the total marks obtained based on all assessments in the class are used as an indicator of the student's achievement. The average marks of each student's peer tutor were calculated and compared with the marks obtained by each student. The difference in marks was obtained between a student and his or her peer tutors and is illustrated in Figure 5. If the difference in marks is more than zero, this indicates that the student

obtained higher marks than the average marks of their peer tutors and vice versa. In other words, the positive difference shows that the student is referring to students who have (on average) higher marks than them while the negative difference shows that a student is referring to students with lower marks (on average) and this may affect their performance.

In the context of this study, students with high weighted indegree (nominated by many students) in the peer tutor network are expected to be high achiever students. However, Figure 4 reveals that a relatively low achiever (node 21) becomes the most referred student as a peer tutor and has many friends. Additionally, this student received lower marks compared to the marks of his or her peer tutors (Figure 5). It is similar to results obtained previously by [16,17] where low achievers become the most referred students in the community. This student is an active learner who loves asking questions and giving responses during online classes. Hence, it is possible that the personality of this student has made him or her popular in the community. Many students nominate this student as a close friend as well as a as peer tutor. However, when a student refers to relative underachievers, this could affect their own understanding, particularly if the peer tutors' understanding of the subject is not that strong. They could transmit false ideas and propagate misconceptions in the online learning community.

### 4.3. Betweenness Centrality and Students' Performance

Students with high betweenness centrality hold the community together. Figure 6 conveys that the betweenness centrality of friendship and the peer tutor network is moderately correlated (coefficient of correlation of 0.5785). This value is lower than the value obtained for weighted indegree in Figure 3. This shows that some students play different roles (as connectors) in friendship and peer tutor networks. It may also reflect the capacity and willingness of the student to teach students from different groups. From the scatter plot, some nodes lie on the x-axis and y-axis. This demonstrates that some students do not act as a connector in the community although we found that all students are connected to each other in this online learning community (from Table 1). The dissemination of knowledge in the online learning community may be hampered by the lack of connection among student groups in the learning community.

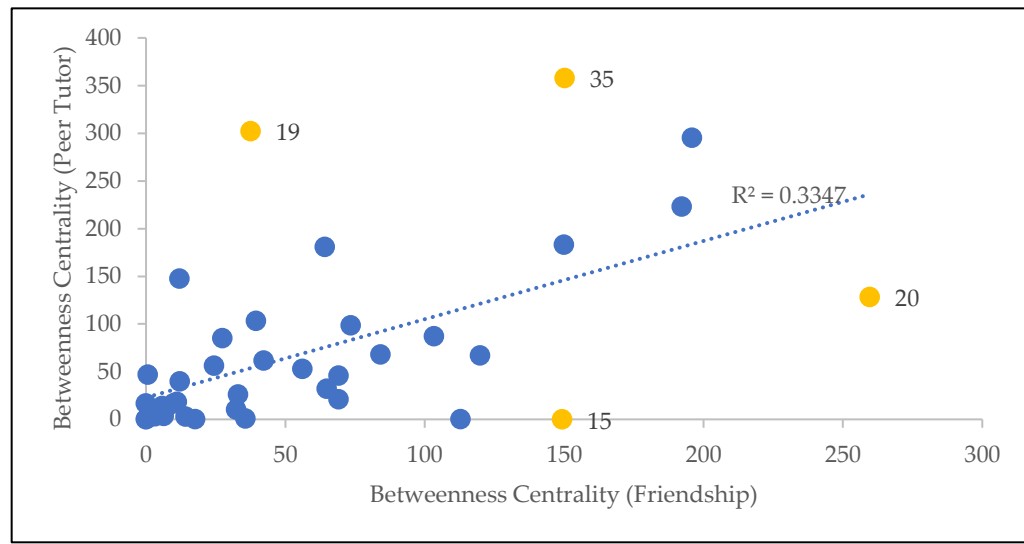

**Figure 6.** The relationship between values of betweenness centrality of the friendship network and betweenness centrality of the peer tutor network. Yellow nodes represent outliers in the scatter plot.

Figure 6 shows four outliers (nodes 15, 19, 20, and 35) colored in yellow. Nodes 19 and 35 represent students who play a significant role in bridging the peer tutor community. Nodes 15 and 20 represent students who are important as a connector much more in the friendship network compared to the peer tutor network. The changes in size of these

outliers can be seen in Figure 7. Based on betweenness centrality, a relatively low achiever student (node 35) becomes the most significant student and acts as a bridge in the peer tutor network according to Figure 7b. Figure 5 reveals that the marks obtained by this student are also far lower than his or her peer tutors' average marks. The distribution of information across the network is impacted by nodes with high betweenness centrality since they are connectors between groups of students. If inaccurate information spreads throughout the network, this scenario could turn out badly as a node with high betweenness centrality can spread information faster compared other nodes. These relationships may have an impact on student performances. Appropriate action should be taken in relation to these nodes in order to avoid the dissemination of the wrong concept within the community.

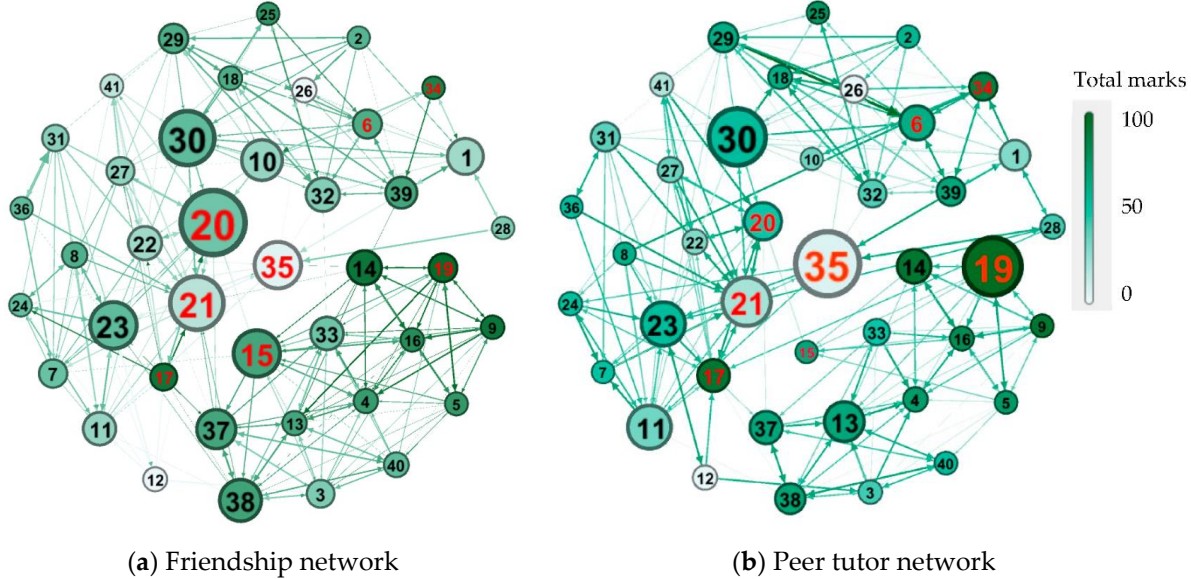

(**a**) Friendship network          (**b**) Peer tutor network

**Figure 7.** The size of the node represents betweenness centrality and its color represents the final marks. Larger nodes represent a higher value of betweenness centrality and darker colored nodes represent higher marks. Red font is used to highlight the nodes that will be discussed in this section.

By comparing Figure 7a,b, nodes 15 and 20 clearly become less important in the peer tutor network as compared to the friendship network. Figure 4 shows that node 15 is less referred to as a peer tutor. These students are moderate students (Figure 4) and obtained marks slightly higher (node 20) and lower (node 15) than the marks of their peer tutor, as illustrated in Figure 5.

Node 19 is the top achiever in the class who is supposed to be a good reference to other students as a peer tutor. In Figure 4, utilizing weighted degree, node 19 seems to not be a popular peer tutor. This student has been referred by six and eight students, respectively, as a peer tutor and close friend. These values are lower than the average in the peer tutor and friendship networks highlighted in Table 1. However, Figures 6 and 7 reveal that this student has a disproportionately high betweenness centrality in the peer tutor networks relative to the amount of his or her friends. This indicates that he or she is referred to by students from a diverse group of students and serves as a main connector in the peer tutoring community of the whole class. This student may help to spread good understanding and knowledge to the community.

## 5. Discussion

The act of social distancing has a positive effect in controlling the spread of the COVID-19 epidemic, however, it also has an isolating effect on students due to the lack of face-to-face interaction during online learning [40], lending to the disintegration of the online learning community. Table 1 conveys that all students were connected to each other during online classes. The result of this study shows a similar result to that obtained by [17] in

a physical class even though for students in our data, only online classes were conducted. This may be due to the fact that this cohort met physically as a class before. Our future research will be on students that started their studies online, having never met some peers physically.

The main purpose of this study is to use social network analysis to understand the formation of online learning communities through interaction between students based on friendship and peer tutor relationships. The results of this study were visualized using directed and weighted networks for both friendship and peer tutors. This study is different from previous studies [16,17] where the study only considers the peer tutor network but does not take into account the underlying friendship network. Figure 3 captures the correlation between the friendship and the peer tutor network in terms of weighted in-degree. It shows that some students are willing and able to teach even though they do not have many close friends. It highlights the peer tutors that educators should be focusing their effort on.

Figure 4a,b shows that node 21 is the most popular student in the class. Figure 7b highlights that node 35 is the main connector in the peer tutor network. Both students act as 'influencers' in the online learning community. However, both these students did not achieve a high mark in this class, in fact, their marks are closer to the bottom half of the class. This may contribute to the sharing of incorrect information with the whole class. By comparing these two students, two questions will be answered. Which of them contributes to the dissemination of knowledge more quickly, as compared to the other? Which 'influencers' in the online learning community are more significant? Refs. [33,34] proved that monitoring a person with high betweenness centrality has a huge impact in slowing the spread of COVID-19. Therefore, by applying the same concept to the dissemination of knowledge, nodes with high betweenness centrality may have a great impact and help accelerate the process of knowledge dissemination in the entire online learning community. With that, node 35 with the highest betweenness centrality is said to be the 'influencer' in the online learning community in this study. Figure 5 reveals that this student scores much lower marks than the marks obtained by his or her peer tutors. Hence, appropriate action should be taken by educators to ensure that this student is communicating the correct information to the entire community. Guidance and assistance can be provided by the educator to improve the understanding of this student which can have a disproportionate effect to the formation of a healthy online learning community.

Alternatively, educators can target other potential students as the main connector in the community. Instead of node 21, Figure 3 reveals that nodes 6, 17, and 34 are significantly important as peer tutors. They are the students who are willing and able to teach other students in the class. Figure 4 illustrates that they are among the high achievers in the community. Moreover, nodes 17 and 34 have obtained significantly higher marks than their peer tutors, as shown in Figure 5. This highlights that they can be a good tutor that can be referred by other students. Hence, the educator can make use of this information to create and form groups that can connect nodes 17 and 34 with the other students who are not close to them. This can foster new engagement in the online learning community. Due to the strong correlation between friendship and peer tutor relationships (Figure 3), this effort will not only help establish new peer tutoring relationships but also foster social friendships. This can also bind the students in this community who are also multi-racial and further help to create a united community in a multi-racial country like Malaysia.

Figure 4 demonstrates that node 19, the class's top achiever, neither becomes the most popular friend nor peer tutor in the classroom. While [12] discovered that students with high achievement are more likely to be referred, this study's findings contradict those findings. Although this student is not the most popular in the community, however, this student becomes one of the important connectors as peer tutors (Figure 6). This helps in the formation of a healthy online learning community because the top student becomes the bridge that holds the learning community together. Additionally, Figure 3 portrays nodes 6, 17, and 34 that have been identified as students who have contributed significantly to the learning community. They teach more than just their friends. Nodes 17 and 34 are high achiever students while node 6 is a moderate student. This study reveals that

these students (nodes 6, 17, 19, and 34) are also important in the development of an online learning environment. They should be highlighted and rewarded by the educators for their contribution to the learning community. Incentive and extra merit can be given to them to encourage them to collaborate more.

As mentioned before, the top achiever (node 19) who is supposed to be a good reference does not seem to have as many friends. Since friendship and peer tutoring are highly correlated, this is not surprising. This highlights the importance of social relationships in the online learning community. This study suggests promoting social interaction between students by setting up an icebreaker so that they can learn more about one another maybe in a playful way. Perhaps using the gamification approach where students can play and learn with their friends at the same time. This approach also demands students' effort to boost their competence, commitment, and social interaction [41]. The study from [42] revealed that scores, scoreboards, badges, and levels are the most popular gamification components to motivate students to compete with each other in a positive way. Indirectly, these elements can be used to foster engagement among students during online classes as well as contribute to data collection. Apart from commonly used platforms such as Google Meet, Microsoft Team, and Zoom, exposure to synchronous virtual discussion platforms such as Gather town can create a sense of virtual learning in the classroom using a video game style [43]. By offering a platform that is playful while learning is being done in accomplishing the assigned activities, it might foster interaction between students, allowing play and learning to be integrated. This may foster a feeling of community in the online learning environment. Students' social interests, such as sharing a hobby or enjoying the same games, can be used to form groups. The result of being able to converse more about the interest they have in common can help students in forging new friendships.

COVID-19 has resulted in changes to the teaching pattern in higher education, and it has also affected the interaction between higher education students. Understanding the nature of the interaction between students during online studies can help in identifying related issues so that various initiatives can be implemented in adapting. The results of this study may be helpful in understanding the interaction between students, to foster a vibrant online learning community.

## 6. Conclusions

In this research, two types of relationships, friendship and peer tutor relationships, were considered to capture students' interaction during the COVID-19 pandemic. The process started with the collection of data using an online questionnaire, followed by the curation of the data to networks using a network analytic tool, Gephi. To ensure that data collection can be done more effectively, certain steps, particularly those requiring the use of technology, and related issues have been highlighted. Using directed and weighted networks for friendship and academic interactions as peer tutors, the data were analyzed and illustrated. This reveals interactions between students and the formation of an online learning community.

The results of the study found that node 21 became the most popular student and node 34 became the main connector in the learning community. Both nodes 21 and 34 are 'influencers' and acts as superspreaders of knowledge in the community. However, both students are relatively low achievers who obtained lower marks than his/her peer tutors. This may contribute to the spread of incorrect information and affect the understanding of the whole learning community. As an alternative, this study has suggested high achievers who are also willing and able to teach (nodes 17 and 34) to be used as main connectors in forming a community with a strong understanding. Interestingly, although the top achiever student (node 19) is not the most popular in the friendship network and peer tutor, this student acts as one of the important connectors in the online learning community. This study was successful in identifying students (nodes 6, 17, 34, and 19) who have contributed significantly to the development of a learning community. They should be rewarded for their contribution in the formation of the online learning community.

This study provides the educator with a broad picture of community dynamics and information sharing across the entire network. Educators can focus their effort to assist and improve the performance of relatively low achievers who are 'influencers' in the community in order to ensure a positive ripple effect throughout the online learning community. To foster new interactions, the educator can assign tasks and form groups to connect students with the students who are not close to them. Incentives can be given to students as a reward for their contribution to the development of the online learning community. This strategy can be used to encourage collaboration and engagement among students and build a healthy learning community.

By comparing friendship and peer tutor relationships, this study reveals the interesting role of various students. This study shows that social network analysis is a powerful tool to understand the formation of an online learning community and further propose strategies to cultivate and foster a thriving online learning community among students in higher education institutions.

**Author Contributions:** Conceptualization, N.Z.A.R., N.N.B., Z.H.Z. and F.A.R.; methodology, N.Z.A.R., N.N.B., N.S.M.A., Z.H.Z., H.B., N.F.M. and F.A.R.; investigation, N.Z.A.R., N.N.B., Z.H.Z. and F.A.R.; validation, N.Z.A.R., N.N.B., N.S.M.A., Z.H.Z., H.B., N.F.M. and F.A.R.; resources, N.Z.A.R., N.N.B., N.S.M.A., Z.H.Z., H.B., N.F.M. and F.A.R.; data curation, N.Z.A.R.; writing—original draft preparation, N.Z.A.R.; writing—review and editing, Z.H.Z. and F.A.R.; visualization, N.Z.A.R.; supervision, Z.H.Z. and F.A.R.; project administration, N.Z.A.R., Z.H.Z. and F.A.R.; funding acquisition, F.A.R. All authors have read and agreed to the published version of the manuscript.

**Funding:** This work was supported by the Universiti Kebangsaan Malaysia (UKM) Internal grants GUP-2021-046 and PDI-2021-030.

**Data Availability Statement:** To preserve the privacy of the students, the data used in this study will not be disclosed.

**Conflicts of Interest:** The authors declare no conflict of interest.

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
