# Peer review of "Comparing Friends and Peer Tutors Amidst COVID-19 Using Social Network Analysis"

_mathematics, doi:10.3390/math11041053_

Round 1

Reviewer 1 Report

The review is attached

Reviewer 2 Report

This article studied the problem of students’ engagement during online learning. The matter is interesting. Social network analysis is a powerful tool to reveal hidden interactions. However, what I am concerned about is the lack of innovation since all the analysis is based on common network measures. I have the following comments on this manuscript:

1. In Sec.1, the motivation of students’ engagement is clear, however, from the title, “Comparing Friends and Peer Tutors” should be the main research content, which is never mentioned in Sec.1. If the authors could explain why “Comparing Friends and Peer Tutors” is significant in Sec.1, that would be better.

2. In Sec.4, visualization results and discussions are good. If the author could add some comparative experiments with existing work, that would be better.

Author Response

Response to Reviewer 2 Comments

Point 1: In Sec. 1, the motivation of students' engagement is clear, however, from the title, "Comparing Friends and Peer Tutors" should be the main research content, which is never mentioned in Sec. 1. If the authors could explain why "Comparing Friends and Peer Tutors" is significant in Sec. 1, that would be better.

Response 1: Thank you for highlighting this matter. We agree that the title “Comparing Friends and Peer Tutors” should be highlighted in Sec. 1. Hence, we have extended Sec. 1 to highlight the importance of comparing friendship and peer tutor relationship. In this Section, we provide motivation from previous research related to different type of relationships. Other studies mostly consider one type of relationship (usually peer tutoring relationship) to understand the formation of learning community. However, in order to fully understand the peer tutoring role in the learning community, we need to take into account other factors influencing students’ interaction including homophily (friends tend to be people with similar attributes) captured through the friendship network. Therefore, we compared friendships and peer tutoring relationships using social network analysis to uncover the ‘real’ peer tutors that teach more than just their friends. Sec. 1 currently spans line (34) to line (92). Here we highlight line (53) to line (92):

“Involvement in friendships and peer tutoring are two crucial forms of engagement that are frequently linked to interactions between students. Both kinds of connections between students develop naturally. Previous studies highlight the importance of friendships amongst higher education students in the learning environment. [13] focuses on the role of friendship in the online learning environment. [14] emphasises the value of developing friendships among students to support their learning and foster an atmosphere where everyone is valued as a source of knowledge. Friends tend to be people with similar attributes (also known as homophily) such as gender and religion. This inevitably impacts the structure of a learning community [15]. Previously in [16,17] we have studied the relationship between peer tutoring and student performance . [17] highlights that peer tutoring does not necessarily correspond with academic success. Therefore, we take into account other variables to further understand peer tutoring and academic performance including homophily which can be captured using friendship networks. “

“This study emphasises social network analysis as the primary technique for examining the dynamics of the formation of an online learning community. A practical solution can be recommended to educators to help monitor and create a healthier learning environment. This study a is continuation of [16,17], further using centrality measures to understand the learning community of Malaysian students. Due to the dearth of studies related to online learning communities, particularly in Malaysia, it is a field that requires attention, especially when dealing with crises such as the COVID-19 pandemic.

Social network analysis is an approach that utilizes data mining to investigate the social structure of a community via graph theory. This strategy has been successfully applied in a variety of disciplines, including finance [18,19], scientific collaboration [20,21], and education [16,17]. In the context of education, this approach can be utilized as a benchmark for monitoring student association through networks and developing solutions that are pertinent to concerns and challenges [22–24]. This enables educators to build a better learning community by ensuring that students’ engagement results in outcomes that are beneficial to the students.

One goal of this study is to encourage researchers, especially educators to utilize social network analysis in order to investigate the structure of online learning communities. The main contribution of this study is to highlight the importance of contrasting friendships and peer tutor relationships to reveal the dynamics of the formation of an online learning community using the social network analysis approach. This study focuses on undergraduate students in Malaysia with online learning experience throughout the whole semester due to movement restrictions during the pandemic COVID-19. To ensure that data are gathered properly, the use of technology and concerns regarding data collection have been taken into consideration. Section 2 discusses relevant literature and Section 3 provides detailed explanations of our research methodology. Findings and analyses are presented in Section 4. Section 5 discusses the findings and offers potential strategies to encourage the formation of a healthy online learning community”

Point 2: In Sec.4, visualization results and discussions are good. If the author could add some comparative experiments with existing work, that would be better.

Response 2: Thank you for pointing this out. We have added some comparative experiments with existing work to support our findings in addition to the previous researches mentioned in point 1. Sec. 4 currently spans line (336) to line (489). Here we highlight some content  

Line (355) to line (366)

Table 1 provides an overview of the friendship and peer tutor network of the 41 students (a whole class) participating in this study. Peer tutor networks have 20% fewer edges than friendship networks. The connected components being 1 indicate that all students are connected in terms of friendships and peer tutors. This means that during the COVID-19 pandemic, students at least have one friend or one peer tutor friend in the class while learning online. This study shows that there is interaction between students during the pandemic, however there may have been a decline as suggested by [40]. Our study shows that all students are still connected in the community, even though online learning have been implemented due to COVID-19. On average, a student has more friends—up to 8 close ones—compared to peer tutors. A student only has 6 peer tutors on average. This shows students of this class engaged less as peer tutors. Lack of engagement among students pertaining academic matters is not surprising.

Line (398) to line (406)

In Figure 4, nodes 17 and 34 are high achievers while node 6 is a moderate achiever. Nodes 17 and 34 obtained significantly higher marks than the marks of their peer tutor (Figure 5). In addition, nodes 17 and 34 were referred to as peer tutors because they were known as excellent students from their past achievements. The results of this study is similar with study from [12] where high achievers become the important students in the learning community. This may be the reason why these students are referred to as peer tutors by other students, although they may have relatively less friends. Node 17 is a good achiever but is from the minority gender. Hence, this might be the reason that this student has not been listed as close friends by many students.

Line (424) to line (436)

In the context of this study, students with high weighted indegree (nominated by many students) in the peer tutor network are expected to be high achiever students. However, Figure 4 reveals that a relatively low achiever (node 21) becomes the most referred student as a peer tutor and has many friends. Additionally, this student received lower marks compared to the marks of his or her peer tutors (Figure 5). It is similar to results obtained previously by [16,17] where low achivers become the most referred students in the community. This student is an active learner who loves asking questions and giving responses during online classes. Hence, it is possible that the personality of this student has made him or her popular in the community. Many students nominate this student as a close friend as well as a as peer tutor. However, when a student refers to relative underachievers, this could affect their own understanding, particularly if the peer tutors’ understanding of the subject is not that strong. They could transmit false ideas and propagate misconceptions in the online learning community.
